# Research on Collaborative Management Strategies of Closed-Loop Supply Chain under the Influence of Big-Data Marketing and Reference Price Effect

**Deqing Ma**  **and Jinsong Hu ***

School of Business, Qingdao University, Qingdao 266071, China; qddxmdq@163.com
* Correspondence: hujinsong@qdu.edu.cn

**Abstract:** This paper integrates the Internet service platform with big-data marketing into the dynamic closed-loop supply chain system as an independent game subject. Considering the reference price effect of consumers, this work constructs differential games among manufacturer, retailer and Internet service platform under three business models of independent, collaborative production and collaborative marketing. Using Bellman's continuous dynamic programming theory, this work obtains the optimal feedback strategies of price and big-data marketing effort, brand goodwill, return rate of used products and corporate profits under the three business models. Comparing the three scenarios and analyzing the sensitivity of key exogenous parameters, it can be found that the involvement of Internet service platform has a crucial impact on the sustainable profitability of supply chain enterprises. Considering the reference price effect of consumers, enterprises should adopt different strategic alliances in different periods, which can also gain new development momentum in the context of data-driven marketing, achieve the improvement of the triple-bottom line of closed-loop supply chain and even reach a win-win situation for supply chain enterprises.

**Keywords:** big-data marketing; reference price effect; closed-loop supply chain; differential game

## 1. Introduction

Closed-loop supply chain (CLSC), as a branch of the supply chain (SC), integrates forward selling process in classical SC, backward activities (e.g., product acquisition, reverse logistics, points of use and disposal, testing, sorting, refurbishing, recovery, recycling) as well as re-marketing and re-selling into a unique system [1–3]. Here, there is both economic and non-economic motivation for focal firms to implement CLSC [4]. As it is reported, the implementation of CLSC can help manufacturers generate 45%–60% cost savings by reusing of used products, reducing at least 300,000 tons of landfill every year [5]. Additionally, the reuse of recycled products reduces the input of new materials and greatly enhances the sustainable development of resources [6]. So more and more manufacturing companies such as Hewlett-Packard, Lenovo, Dell, Xerox and other digital products manufacturing, as well as automobile manufacturing such as BMW and Volvo, have adopted the closed-loop supply chain as their corporate development strategy and they did achieve it in practice. For example, Xerox has recycled more than 60% of its cartridges in Europe and North America, gaining a higher competitive advantage and earning more profit and greater market share. Volvo S40, designed and manufactured by Volvo, about 85% of the entire car can be recycled and reused. Plastic, felt-containing and wood-based interior materials can be recycled and reused, reducing the use of PVC in the entire vehicle manufacturing process, which greatly reduces the health hazard to workers and people in the area around the factory and improves the corporate social image. Due to its triple-bottom line of economic, social and environmental [7], CLSC has shown far more advantages than the traditional SC.

Therefore, the management of CLSC has also become a focus in the theoretical and practical circles. In addition, due to the inherent dynamic characteristics of the recycling process and the impact of waste recycling on the future performance of the company, the theoretical community is increasingly incorporating the CLSC into the dynamic research perspective, and developing a CLSC with dynamic optimization and differential game theory [8].

In recent years, new digital marketing channels based on e-commerce and social media have emerged, data has exploded, and more and more companies are beginning to recognize the value of big data [9–12]. It is pointed out that the effective use of big data can help companies make better decisions [13,14]; data analysis companies and third-party Internet service platforms came into being, which also revolutionized closed-loop supply chain management. More and more companies choose to cooperate with third-party Internet service platforms in order to obtain higher-quality user information and understand and predict customer needs more accurately [15]. Unilever, a publicly traded company that sells food and laundry products, even created a new marketing executive position, Big-Data Marketing Officer, to drive corporate marketing reform [16].Research shows that with big-data marketing technology, closed-loop supply chain companies can better analyze user behavior, increase product exposure, enhance brand goodwill (It refers to the comprehensive image of a brand in the minds of consumers—including its attributes, quality, grade (taste), culture, personality, etc., and can be considered a significant asset of a firm [17]), and enhance consumer environmental awareness, effectively improving the efficiency of closed-loop supply chain operations [9,18]. For example, Lenovo, Huawei, Apple and other companies cooperate with Internet service platforms such as JD (a self-run e-commerce enterprise in China) and Amazon. These platforms can conduct big-data marketing for potential customers, increase consumer conversion rate, and increase brand goodwill of the products they sell by analyzing users' browsing records, shopping cart addition records, consumption records and even GPS positioning data [19,20].

It can be seen that the formulation of any business strategy in a closed-loop supply chain is inseparable from the consideration of consumer behavior. Some research even defines marketing as consumer-oriented science [12,21], that is, only by truly understanding the hearts and needs of consumers can enterprises make more realistic and competitive decisions. In addition to the product of the new era of big-data marketing, the price is always an important decision in the sales process of the enterprise, and the price cannot be viewed in isolation. Consumers often compare product expectations (i.e., reference prices) with actual product prices to make their purchasing decisions, which are referred to the reference price effect in consumer behavior [22]. The definition of reference price in the theoretical circle is mainly divided into two categories: one class thinks that the consumer forms a memory of the price based on the past purchase experience, which is called the Internal Reference Price (IRP); the others believe that the reference price is formed in the consumption process, based on external factors such as quality, brand goodwill, etc. called External Reference Prices (ERP) [23]. By comparing the ERP and IRP in specific contexts, Hardy et al. [24] found that the reference price based on factors of quality and brand goodwill is suitable for most situations. Surprisingly, most of the existing studies on the reference price effect take IRP as the reference [25,26]. In this case, it is imperative to explore the influence of ERP on enterprises, especially for consumer-oriented businesses.

To this end, this paper will build a closed-loop supply chain system consisting of a manufacturer engaged in recycling and remanufacturing, a retailer and an Internet service platform. Taking into account the external reference price effect of consumers and big-data marketing of Internet service platform, this work constructs three different business modes between channel members, that is, independent operation model, collaborative production model and collaborative marketing model. By solving and discussing the three different business models, the following three questions are going to be answered:

(1) What is the wholesale price strategy of manufacturer, retail price strategy of retailer and big-data marketing effort of Internet service platform, as well as brand goodwill, return rate and supply chain profit under the three business models?

(2)　How does the consumer reference price effect affect the economic, environmental and social performance of supply chain enterprises?

(3)　In different operating periods, what operation mode should enterprises adopt to achieve the maximum of economic, social and environmental benefits—independent operation, production alliance between retailers and manufacturers or marketing alliance between retailers and Internet service platforms?

Our main results are as follows. First, all strategies and performance of CLSC members under three business models are related to the brand goodwill, so improving the brand goodwill is an important way for each company to enhance profit. Second, when the reference price effect is small, the retailer will reduce retail prices and mitigate the double marginal effect of the supply chain under the collaborative production and collaborative marketing business model; the big-data marketing efforts under the independent operation and collaborative production mode will follow with the increase of the reference price effect; the big-data marketing effort will exist only when the reference price effect factor is within a certain range. What is more, the application of big-data marketing enables enterprises to transform marketing activities into customer-oriented science, and firms should take strategic alliance in different periods to realize the triple-bottom-line of closed-loop supply chain and reach a win-win situation for supply chain enterprises.

The three areas related to this study are: (1) dynamic closed-loop supply chain management; (2) big-data marketing; and (3) consumer reference price effects. In order to sort out the research related to this paper and highlight the contribution of this paper, the more representative articles in the three fields are listed in Table 1 and compared and analyzed below.

In recent decades, most studies focusing on CLSC have been developed from the perspective of static category [4,5,27–30], but due to the inherent dynamic characteristics of the recycling process, it is worth pointing out that it is a consensus of the theoretical community to use the optimal control theory and differential game theory to study the CLSC [4,6–8]. Most of the researches on the existing dynamic closed-loop supply chain management focus on optimal decision-making problems of enterprises [7,31,32], recycling channel selection [33], contract mechanism designing [6,7] and the influence of member behavior on decision-making and performance of CSLC [34,35]. These studies have proved the necessity of studying closed-loop supply chains from a dynamic perspective, and the effectiveness of cooperation among CSLC companies to promote economic and environmental performance. However, it is surprising that none of these studies started with consumers and did not include their decision-making behaviors in the shopping process into the influencing factors of companies' operational strategies, which have been proven to be the key for companies to acquire consumers and gain a competitive advantage in the market [12,21]. To this end, this article incorporates the reference price behavior that is prevalent in the consumer shopping process into considerations for corporate decision-making.

In the study of reference price effects, existing research uses the internal reference price model (IRP) to describe the formation of consumer reference prices [36–40]. Among them, Zhang et al. [36] found that the reference price effect always has a positive impact on sales. Zhang et al. [37] found that regardless of the way companies collaborate, the supply chain will have better economic benefits when consumers have higher initial reference prices, are more sensitive to reference price effects, and are more loyal to brands. Lu et al. [38] have also found that reference price effect has a great effect on the optimal marketing strategy and consumer demand of the monopoly. Xu et al. [39] explored the optimal recycling channel selection problem regarding IRP effect, and found that the higher IRP effect would reduce the profit of both manufacturers and retailers. Zhou et al. [39] further analyzed the conditions for enterprises to adopt price commitment or advertising commitment under IRP effect. It can be found that most of the existing research based on the reference price effect, on the one hand, is concentrated in the forward marketing supply chain, which is less involved in the recycling and remanufacturing of used products; on the other hand, almost all existing researches use the consumer's IRP effect, and its actual usability is not as good as external reference price (ERP) effect, which has

been proven to be closer to consumers' shopping preferences [24]. Therefore, with the help of ERP, this paper will better integrate the consumer's reference price behavior into the decision-making process of enterprises and discuss the operation strategies and cooperation plans of CLSC members based on the consumer's decision-making behavior.

**Table 1.** Comparison of some important literatures.

| Author | Dynamic Closed-Supply Chain | | | Big-Data Marketing | Reference Price Effect | Cooperation among Enterprises |
| --- | --- | --- | --- | --- | --- | --- |
| | Economic | Environment | Society | | IRP/ERP | |
| Giovanni et al. [6] | √ | √ | | | | √ |
| Hong et al. [31] | √ | √ | | | | √ |
| He [31] | √ | √ | √ | | | √ |
| Huang et al. [32] | √ | √ | | | | √ |
| Ma et al. [34] | √ | √ | | | | |
| Giovanni [7] | √ | √ | √ | | | √ |
| Xiao et al. [35] | √ | √ | | | | |
| Zhang et al. [36] | | | | | IRP | √ |
| Zhang et al. [37] | | | | | IRP | |
| Lu et al. [38] | √ | √ | | | IRP | |
| Xu et al. [39] | √ | √ | | | IRP | |
| Zhou et al. [40] | | | | | IRP | |
| Wu et al. [19] | √ | | | √ | | |
| Xiang et al. [41] | √ | | | √ | | |
| This paper | √ | √ | √ | √ | ERP | √ |

As a new marketing method under the large-scale explosion of data, big-data marketing saves unnecessary inventory costs for enterprises due to its precise insights and personalized marketing methods for consumers, achieves accurate marketing, and enhances the market advantage of enterprises. It is now widely used in practice, while in the theoretical community, there is very little research on big-data marketing, not to mention in the study of CLSC. Wu et al. [19] studied the supply chain operation strategy of big-data service providers as an independent entity based on differential game theory. They found that under the background of big-data marketing, the synergy between enterprises can achieve a win-win situation for supply chain enterprises. Xiang et al. [41] studied the dynamic R&D and advertising investment strategies of closed-loop supply chain enterprises under the background of big-data marketing, and found out the free-riding behavior of retailers when manufacturers bear part of the big-data marketing costs. Obviously, since the CLSC promotes economic, environmental and social benefits, the application of big-data marketing technology in its operation process is particularly worthy of study and discussion.

Based on the related research in the above three fields, this paper considers the external reference price effect of consumers in the context of big-data marketing, and compares the optimal marketing strategies under the different synergistic differential game mode among manufacturer, retailer and Internet service platform. We also determine the optimal business model of enterprises to achieve the triple benefits of CLSC. The main contributions of this paper are: 1) further enrich the research results of the dynamic CLSC and expand existing research in the influence of ERP in consumers' shopping process; 2) incorporate Internet service platform as an independent game subject into the CLSC and use its big-data marketing as a new marketing method; 3) and further explore the optimal vertical cooperation between enterprises to achieve economic, environmental and social benefits of CLSC.

The remainder of this paper is organized as follows: Section 2 gives the construction of model and the corresponding hypothesis. Section 3 resolves the feedback strategies and performance of CLSC

members in the model. Section 4 is the analytical comparation while Section 5 is the numerical analysis. Section 6 concludes the paper.

## 2. The Models

Consider a three-stage closed-loop supply chain system consisting of one manufacturer *M*, one retailer *R*, and one Internet service platform *I*, in which *M* is responsible for recycling used products and remanufacturing activities, and determines the wholesale price $w(t)$ of the product. *R* purchases goods from *M*, determining the retail price $p(t)$ of the product and sells to consumers. In order to accurately locate the customer group in the sales process, and promote consumer demand, *R* will use *I* to conduct big-data marketing, and *I* determines its big-data marketing effort which includes estimating consumer preferences through data mining, cloud computing and other technologies for consumers' past consumption records, and recommending products to consumers to improve consumer demand. These three form a *M*-led Stackelberg differential game. The relevant assumptions related to this article are as follows:

Assume that the Internet service platform can increase the exposure of retailers to sell products through accurate customer positioning and personalized product recommendation, and makes consumers know more about a brand to enhance the brand goodwill in time. Using the Nerlove-Arrow (N-A) model [42] to characterize the impact of big-data marketing efforts on brand goodwill, the differential equation for the change in brand goodwill can be expressed as:

$$\dot{G}(t) = \gamma B(t) - \delta G(t), G(0) = G_0 > 0 \tag{1}$$

In Equation (1), $\gamma = 0$ represents the efficiency of big-data marketing efforts to improve brand goodwill. In addition, due to competition between brands and consumers' forgetting of brands, brand goodwill is exponentially attenuated by the speed of $\delta = 0$. It is worth pointing out that although the N-A model is used here to describe the goodwill dynamics, the marketing method of advertising has been replaced by big-data marketing. The reason is that traditional marketing methods are based on small data. The lag and limitation are obvious, and due to the diversity of consumer demand for products and the timeliness as well as the multi-platformness of their shopping decisions, based on big-data analysis, firms can accurately predict consumer demand and implement personalized and precise marketing. The superiority of big-data marketing technology is evident.

The return rate of used products is always positively related to brand goodwill, that is, the higher the brand's goodwill, the easier it is to motivate consumers to return used products and promote remanufacturing engineering [8,41]. Therefore, the return rate $\tau(t)$ can be expressed as:

$$\tau(t) = \rho G(t) \tag{2}$$

where $\rho > 0$ is the correlation coefficient between return rate and goodwill, meanwhile, $\rho$ can be seen as an adjusting parameter to ensure the return rate $\tau(t) \in [0,1], t \in [0, +\infty)$, that is, the percentage of used product return from consumers.

Assume that the consumer will make an expected judgment on the product price based on the brand goodwill before purchasing the product, which is called the reference price [4], i.e.,

$$R_p(t) = \mu G(t) \tag{3}$$

where $\mu > 0$ represents the correlation coefficient between the reference price $R_p(t)$ and the brand goodwill $G(t)$. Equation (3) indicates that people always expect that a product with a high brand goodwill is always priced higher.

Due to the existence of the reference price, the reference price effect becomes an important factor affecting the consumer's decision to make a purchase: when the reference price $R_p(t)$ is higher than the actual retail price $p(t)$ of the product, the consumer obtains the consumer surplus, and the purchase

decision will be reached. The demand for the brand is increased. Conversely, when the reference price $R_p(t)$ is lower than the actual retail price $p(t)$ of the product, the consumer chooses to abandon the purchase of the branded product or switch to another brand, which will result in a decrease in the demand for the branded product in the consumer market [35,36]. Therefore, the reference price $R_p(t)$ can also be considered as consumers' highest willingness to pay for the product. At the same time, consumers' demand will increased with return rates, which can be explained by the fact that when people choose to return used products, they often choose to purchase new products to meet the continued demand for the product. In addition, products with high brand goodwill will always bring more consumer demand [7,8]. Therefore, the demand function of the consumer market can be expressed as:

$$D(t) = \beta\big[R_p(t) - p(t)\big] + \varepsilon\tau(t) + \theta G(t) \tag{4}$$

where $\beta > 0$ is the impact factor of the reference price effect on demand, and $\varepsilon, \theta > 0$ represent the impact of return rate and brand goodwill on demand, respectively.

Substituting Equations (2) and (3) into Equation (4), the requirement function can be written as:

$$D(t) = \Phi G(t) - \beta p(t) > 0 \tag{5}$$

where $\Phi = \varepsilon\rho + \beta\mu + \theta$.

There are two main sources of marginal revenue for manufacturers: one is the marginal revenue of product wholesale $w(t)$, and the other is the marginal revenue of recycling and remanufacturing $(\Delta - c_T - c_M)\tau(t)$, where $\Delta, c_T, c_M > 0$ represent the marginal residual value of used products, the marginal transfer cost of used products from consumers to manufacturer, and the marginal production cost of remanufacturing, assuming $\Delta - c_T - c_M > 0$, that is, manufacturers are always profitable in recycling and remanufacturing activities, remanufacturing projects can continue to develop, and as a profit center for manufacturer's production [8,41]. Therefore, the marginal benefit of a manufacturer's production activity can be expressed as $w(t) + KG(t)$, where $K = \rho(\Delta - c_T - c_M)$.

The marginal revenue of $R$ is mainly derived from the sales activity of the products, i.e., $p(t) - w(t)$. In addition, $R$ will use $I$ for big-data marketing to consumers and charged by $I$ with the service fee $\xi D(t)$ [9,41], where $\xi > 0$ represents the unit service rate, and $\xi > 0$ is a constant. In order to ensure the normal operation of retailers, it is necessary to set up $p(t) - w(t) - \xi > 0$. The marginal benefit of $I$ is $\xi$, and the big-data marketing cost is assumed to be $\frac{kB^2(t)}{2}$, where $k > 0$ is the cost factor, which satisfies the law of increasing marginal costs [19,41].

In summary, it is assumed that $M$, $R$ and $I$ each make their own profit maximization within the unlimited planning period, and their target functionals can be expressed as:

$$\max_{w(\cdot)}\Big\{J_M = \int_0^{+\infty} \big\{e^{-rt}[w(t) + KG(t)][\Phi G(t) - \beta p(t)]\big\}dt\Big\}$$

$$\max_{p(\cdot)}\Big\{J_R = \int_0^{+\infty} e^{-rt}\{[p(t) - w(t) - \xi][\Phi G(t) - \beta p(t)]\}dt\Big\}$$

$$\max_{B(\cdot)}\Big\{J_I = \int_0^{+\infty} e^{-rt}\Big\{\xi[\Phi G(t) - \beta p(t)] - \frac{kB^2(t)}{2}\Big\}dt\Big\}$$

where $r > 0$ is the discount factor.

## 3. Feedback Stackelberg Equilibria and Steady States

In this section, we seek to derive the models analytically and explore the feedback Stackelberg equilibria and steady states of manufacturer, retailer and Internet service platform under three scenarios: independent operation scenario (model $N$), cooperative production scenario (model $P$), and cooperative marketing scenario (model $S$). The business model in SC are shown in Figure 1.

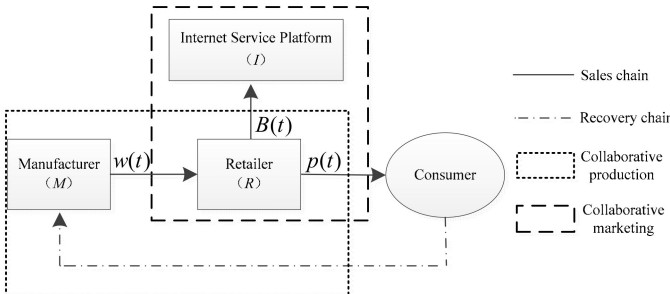

**Figure 1.** Business Model in SC.

In order to express the model and make the analysis clear, three scenarios are distinguished by superscript *N*, *P* and *S*, which represents independent operation, collaborative production and collaborative marketing, respectively. What is more, we use subscript *M*, *R* and *I* to denote manufacturer, retailer and Internet service platform, and subscript *MR* and *RI* for collaborative production alliance and collaborative marketing alliance.

### 3.1. Independent Business Scenario (Model N)

Under the independent business model, manufacturer, retailer and Internet service platform seek to pursue the maximization of their own profit, and carry out Stackelberg differential game, in which *M* plays as the dominant while others as the followers in the planning period. The order of game is as follows: *M* announces his wholesale price $w^N(t) > 0$, then *R* reads the announcement and determines his retail price $p^N(t) > 0$. Meanwhile, in the process of sales, in order to accurately locate consumer groups, *R* will use the big-data marketing service of *I* who determines his big-data marketing efforts. The differential game model can be summarized as:

$$\max_{w^N(\cdot)}\left\{J_M^N = \int_0^{+\infty} e^{-rt}\left[w^N(t) + KG(t)\right]\left[\Phi G(t) - \beta p^N(t)\right]dt\right\}$$

$$\text{s.t.}\begin{cases} \max_{p^N(\cdot)}\left\{J_R^N = \int_0^{+\infty} e^{-rt}\left\{\left[p^N(t) - w^N(t) - \xi\right]\left[\Phi G(t) - \beta p^N(t)\right]\right\}dt\right\} \\ \max_{B^N(\cdot)}\left\{J_I^N = \int_0^{+\infty} e^{-rt}\left\{\xi\left[\Phi G(t) - \beta p^N(t)\right] - \frac{k\left(B^N(t)\right)^2}{2}\right\}dt\right\} \\ \dot{G}(t) = \gamma B^N(t) - \delta G(t), G(0) = G_0 > 0 \end{cases}$$

**Proposition 1.** Under the independent business model, feedback strategies of *M*'s optimal wholesale price, *R*'s optimal retail price, and *I*'s optimal big-data marketing effort are:

$$w^N = \frac{1}{2\beta}\left[(\Phi - \beta K)G^N - \beta\xi\right], p^N = \frac{1}{4\beta}\left[(3\Phi - \beta K)G^N + \beta\xi\right], B^N = \frac{\xi\gamma(\Phi + \beta K)}{4(r+\delta)k}.$$

The firms' optimal value function, which can also be seen as their profits over time, are given by:

$$V_M^N = \frac{(\Phi + \beta K)^2}{8(r+2\delta)\beta}\left(G^N(t)\right)^2 + \left[\frac{\xi\gamma^2(\Phi + \beta K)^3}{16(r+\delta)^2(r+2\delta)\beta k} - \frac{\xi(\Phi + \beta K)}{4(r+\delta)}\right]G^N(t) + \frac{\beta\xi^2}{8r} + \frac{\gamma^2\xi(\Phi + \beta K)}{4r(r+\delta)k}\left[\frac{\xi\gamma^2(\Phi + \beta K)^3}{16(r+\delta)^2(r+2\delta)\beta k} - \frac{\xi(\Phi + \beta K)}{4(r+\delta)}\right]$$

$$V_R^N = \frac{(\Phi + \beta K)^2}{16(r+2\delta)\beta}\left(G^N(t)\right)^2 + \left[\frac{\xi\gamma^2(\Phi + \beta K)^3}{32(r+\delta)^2(r+2\delta)\beta k} - \frac{\xi(\Phi + \beta K)}{8(r+\delta)}\right]G^N(t) + \frac{\beta\xi^2}{16r} + \frac{\gamma^2\xi(\Phi + \beta K)}{4r(r+\delta)k}\left[\frac{\xi\gamma^2(\Phi + \beta K)^3}{32(r+\delta)^2(r+2\delta)\beta k} - \frac{\xi(\Phi + \beta K)}{8(r+\delta)}\right]$$

$$V_I^N = \frac{\xi(\Phi + \beta K)}{4(r+\delta)}G^N(t) + \frac{\gamma^2\xi^2(\Phi + \beta K)^2}{32r(r+\delta)^2k} - \frac{\beta\xi^2}{4r}$$

with the optimal time evolution trajectory of $G^N(t) = e^{-\delta t}\left[G_0 - \frac{\xi\gamma^2(\Phi + \beta K)}{4\delta(r+\delta)k}\right] + \frac{\xi\gamma^2(\Phi + \beta K)}{4\delta(r+\delta)k}$.

**Proof.** See the Appendix A.

Undoubtedly, both *M*'s wholesale price and *R*'s retail price are dynamic strategies which are adjusted over time and always proportional to the brand goodwill of products. This shows that the promotion of brand goodwill enables *M* or *R* to wholesale or sell products at higher prices and obtain

more production or sale incentives. On the other hand, wholesale price and retail price will also aggravate the double marginal effects of the supply chain system. Thus, the higher brand goodwill of products that consumers want to get, the more they have to pay.

As for $I$, his big-data marketing effort, $B^N$, does not change over time under the independent business scenario, and has no obvious relationship with the brand goodwill. But the it is analyzed that $B^N$ is positively correlated to the impact factor of goodwill, $\gamma$, which indicates that the greater the impact of marketing effort on goodwill, the more motivation $I$ will get to pay for big-data marketing effort. The reason behind which is that big-data marketing effort is a key factor to enhance brand goodwill and further stimulate demand. What's more, with certain marginal revenue, higher demand will make $I$ more profitable.

Under the independent business model, the profits of $M$, $R$ and $I$ are positively related to brand goodwill, which means that in order to achieve higher economic performance, one of the most critical methods is improving brand goodwill, that should be the common goal of all enterprises in the supply chain.

**Corollary 1.** Under the independent business model, the steady states of $w^N$, $p^N$ and $B^N$ are

$w_\infty^N = \frac{(\Phi - \beta K)G_\infty^N - \beta\xi}{2\beta}$, $p_\infty^N = \frac{(3\Phi - \beta K)G_\infty^N + \beta\xi}{4\beta}$, $B^N = \frac{\xi\gamma(\Phi + \beta K)}{4(r+\delta)k}$ respectively.

The steady states of their optimal value functions are:

$$V_{M\infty}^N = \frac{(\Phi + \beta K)^2}{8(r+2\delta)\beta}\left(G_\infty^N\right)^2 + \left[\frac{\xi\gamma^2(\Phi + \beta K)^3}{16(r+\delta)^2(r+2\delta)\beta k} - \frac{\xi(\Phi + \beta K)}{4(r+\delta)}\right]G_\infty^N + \frac{\beta\xi^2}{8r} + \frac{\gamma^2\xi(\Phi + \beta K)}{4r(r+\delta)k}\left[\frac{\xi\gamma^2(\Phi + \beta K)^3}{16(r+\delta)^2(r+2\delta)\beta k} - \frac{\xi(\Phi + \beta K)}{4(r+\delta)}\right],$$

$$V_{R\infty}^N = \frac{(\Phi + \beta K)^2}{16(r+2\delta)\beta}\left(G_\infty^N\right)^2 + \left[\frac{\xi\gamma^2(\Phi + \beta K)^3}{32(r+\delta)^2(r+2\delta)\beta k} - \frac{\xi(\Phi + \beta K)}{8(r+\delta)}\right]G_\infty^N + \frac{\beta\xi^2}{16r} + \frac{\gamma^2\xi(\Phi + \beta K)}{4r(r+\delta)k}\left[\frac{\xi\gamma^2(\Phi + \beta K)^3}{32(r+\delta)^2(r+2\delta)\beta k} - \frac{\xi(\Phi + \beta K)}{8(r+\delta)}\right],$$

$$V_{I\infty}^N = \frac{\xi(\Phi + \beta K)}{4(r+\delta)}G_\infty^N + \frac{\gamma^2\xi^2(\Phi + \beta K)^2}{32r(r+\delta)^2 k} - \frac{\beta\xi^2}{4r}.$$

where the brand goodwill at the stable steady state is $G_\infty^N = \frac{\xi\gamma^2(\Phi + \beta K)}{4\delta(r+\delta)k}$.

According to the optimal brand goodwill function in Proposition 1, as $\delta > 0$, so it is clear to draw the conclusion that brand goodwill is globally stable. This means that the above result is the optimal equilibrium under model $N$, which has important guiding significance for the actual operation of enterprises.

*3.2. Collaborative Production Scenario (Model P)*

Under the collaborative production scenario, $M$ and $R$ cooperate with each other to form the alliance $MR$, jointly complete the process of the recycling, production and sales, and determine the retail price $p^P(t)$ of the product, while $I$ determines its big-data marketing effort, each of which seeks to maximize its own profit, that is, $MR$ and $I$ conduct Nash non-cooperative game. The reason for this joint production mode is that with the help of the Internet service platform, retailers that are directly facing consumers can more accurately target consumers through big-data marketing [41]. According to the individual needs of consumers, retailers can cooperate with manufacturers to produce products that meet consumer needs, avoiding product bias caused by blind guessing of consumer preferences based on small data in classic marketing, and avoiding inventory backlog caused by inaccurate market demand prediction. The differential game model under this condition is:

$$\max_{p^P(\cdot)}\left\{J_{MR}^P = \int_0^{+\infty} e^{-rt}\left\{\left[p^P(t) + KG(t) - \xi\right]\left[\Phi G(t) - \beta p^P(t)\right]\right\}dt\right\}$$

$$\max_{B^P(\cdot)}\left\{J_I^P = \int_0^{+\infty} e^{-rt}\left\{\xi\left[\Phi G(t) - \beta p^P(t)\right] - \frac{k\left(B^P(t)\right)^2}{2}\right\}dt\right\}$$

$$\text{s.t. } \dot{G}(t) = \gamma B^P(t) - \delta G(t), G(0) = G_0 > 0$$

**Proposition 2.** Under the collaborative production scenario, the feedback retail price strategy of $MR$ 's and $I$ 's optimal big-data marketing effort are: $p^P(t) = \frac{(\Phi - \beta K)G^P(t) + \beta\xi}{2\beta}$, $B^P = \frac{\gamma\xi(\Phi + \beta K)}{2k(r+\delta)}$.

Their optimal value functions are:

$$V_{MR}^P = \frac{(\Phi+\beta K)^2}{4(r+2\delta)\beta}\left(G^P(t)\right)^2 + \left[\frac{\gamma^2\xi(\Phi+\beta K)^3}{4(r+\delta)^2(r+2\delta)\beta k} - \frac{\xi(\Phi+\beta K)}{2(r+\delta)}\right]G^P(t) + \frac{\beta\xi^2}{4r} + \frac{\xi\gamma^2(\Phi+\beta K)}{2r(r+\delta)k}$$

$$\left[\frac{\gamma^2\xi(\Phi+\beta K)^3}{4(r+\delta)^2(r+2\delta)\beta k} - \frac{\xi(\Phi+\beta K)}{2(r+\delta)}\right]$$

$$V_I^P = \frac{\xi(\Phi+\beta K)}{2(r+\delta)}G^P(t) + \frac{\xi^2\gamma^2(\Phi+\beta K)^2}{4r(r+\delta)^2 k} - \frac{\beta\xi^2}{2r}$$

with $G^P(t) = e^{-\delta t}\left[G_0 - \frac{\xi\gamma^2(\Phi+\beta K)}{2(r+\delta)\delta k}\right] + \frac{\xi\gamma^2(\Phi+\beta K)}{2(r+\delta)\delta k}$.

**Proof.** See the Appendix A.

As Proposition 2 represents, in the collaborative production scenario, *M* and *R* reach a synergistic production alliance *MR*. And the marginal residual value of the recycled used products belongs to the alliance, eliminating the wholesale marginal cost $w(t)$ of *R* that he should pay under the independent business mode. Besides, it's also avoiding the secondary markup generated by the wholesale trade of goods between *M* and *R* which reduces the double marginal effect of the supply chain. The optimal retail price $p^P(t)$ of the collaborative production alliance is still a time-varying strategy and is directly proportional to the brand goodwill.

Under the collaborative production mode, *I* still maintains independent operation, and its optimal big-data marketing effort strategy is still unchanged, only related to various external environmental factors. However, compared with the independent business model *N*, the big-data marketing efforts under the collaborative production model are increased. The reason is that the collaborative production mode reduces the double marginal effect of the supply chain, stimulates the demand of the consumer market, and makes *I* more power to conduct big-data marketing and gain more profit.

The profit of *MR* and *I* are positively related to brand goodwill. It can be seen that whether the supply chain enterprise is independent or partial alliance, it should pay attention to the positive effect brought by the promotion of brand goodwill to enterprise performance.

**Corollary 2.** Under the collaborative production scenario, the steady states of *MR*'s optimal retail price and *I*'s big-data marketing effort are $p_\infty^P = \frac{(\Phi-\beta K)G_\infty^P + \beta\xi}{2\beta}$.

The steady states of their optimal value functions are:

$$V_{MR\infty}^P = \frac{(\Phi+\beta K)^2}{4(r+2\delta)\beta}\left(G_\infty^P\right)^2 + \left[\frac{\gamma^2\xi(\Phi+\beta K)^3}{4(r+\delta)^2(r+2\delta)\beta k} - \frac{\xi(\Phi+\beta K)}{2(r+\delta)}\right]G_\infty^P + \frac{\beta\xi^2}{4r} + \frac{\xi\gamma^2(\Phi+\beta K)}{2r(r+\delta)k}$$

$$\left[\frac{\gamma^2\xi(\Phi+\beta K)^3}{4(r+\delta)^2(r+2\delta)\beta k} - \frac{\xi(\Phi+\beta K)}{2(r+\delta)}\right],$$

$$V_{I\infty}^P = \frac{\xi(\Phi+\beta K)}{2(r+\delta)}G_\infty^P + \frac{\xi^2\gamma^2(\Phi+\beta K)^2}{4r(r+\delta)^2 k} - \frac{\beta\xi^2}{2r}.$$

where the brand goodwill at the stable steady state is $G_\infty^P = \frac{\xi\gamma^2(\Phi+\beta K)}{2\delta(r+\delta)k}$.

As the proof is similar to the proof of Proposition 1, it will not be repeated here.

*3.3. Collaborative Marketing Scenario (Model S)*

This business model is adopted by some powerful companies under big-data-driven marketing. They generally have sufficient funds and manpower, and also have enough customer data to be able to carry out big-data marketing independently. Such as, Unilever independently established a digital marketing department [16] and JD built a precision marketing framework [20], all of which aim to use the new technology of big-data marketing to more accurately put products into the hands of customers and enhance their shopping experience. Under the collaborative marketing business model, *M*, as the leader, determines the wholesale price $w^s(t)$ of its products, *R* and *I* form a marketing alliance, *RI*, who purchase products from *M*, jointly implement big-data marketing services and retail process, determine big-data marketing effort $B^s(t)$ and products' retail price $p^s(t)$. *M* and *RI* adopt a *M*-led Stackelberg differential game as:

$$\max_{w^S(\cdot)}\left\{ J_M^S = \int_0^{+\infty} \left\{ e^{-rt}\left[ w^S(t) + KG(t) \right]\left[ \Phi G(t) - \beta p^S(t) \right] \right\}dt \right\}$$

$$\text{s.t.}\begin{cases} \max_{p^S(\cdot),B^S(\cdot)}\left\{ J_{RI}^S = \int_0^{+\infty} e^{-rt}\left\{ \left[ p^S(t) - w^S(t) \right]\left[ \Phi G(t) - \beta p^S(t) \right] - \frac{k\left(B^S(t)\right)^2}{2} \right\}dt \right\} \\ \dot{G}(t) = \gamma B^S(t) - \delta G(t), G(0) = G_0 > 0 \end{cases}$$

**Proposition 3.** Under the collaborative marketing business model, *M*'s optimal product wholesale price, the collaborative marketing alliance's optimal big-data marketing effort and the product retail price

are: $w^S(t) = \frac{(\Phi - \beta K)G^S(t)}{2\beta}, p^S(t) = \frac{(3\Phi - \beta K)G^S(t)}{4\beta}, B^S(t) = \frac{1}{2\gamma}\left[ r + 2\delta - \sqrt{(r+2\delta)^2 - \frac{\gamma^2(\Phi+\beta K)^2}{2\beta k}} \right]G^S(t).$

The optimal value functions of *M* and *RI* are:

$$V_M^S = \left[ (r+2\delta)^2 - \frac{\gamma^2(\Phi+\beta K)^2}{2\beta k} \right]^{-1/2}\frac{(\Phi+\beta K)^2}{8\beta}\left( G^S(t) \right)^2,$$

$$V_{RI}^S = \frac{k}{4\gamma^2}\left[ r + 2\delta - \left( (r+2\delta)^2 - \frac{\gamma^2(\Phi+\beta K)^2}{2\beta k} \right)^{1/2} \right]\left( G^S(t) \right)^2.$$

with the time evolution trajectory of

$$G^S(t) = e^{\left[ \frac{1}{2}r - \frac{1}{2}\left( (r+2\delta)^2 - \frac{\gamma^2(\Phi+\beta K)^2}{2\beta k} \right)^{1/2} \right]t}G_0.$$

Under the collaborative marketing business scenario *S*, *R* and *I* form a collaborative marketing alliance *RI* to jointly develop big-data marketing effort and product retail price, while *M* remains independent operation. Under this business model, not only *M* 's product wholesale price $w^S(t)$ and the retail price $p^S(t)$ of *RI* are time-varying strategies, but also the big-data marketing effort strategy $B^S(t)$ is different from the other two business models. It is clear that the optimal decision of the enterprise under this business model is the feedback strategy of goodwill, and all three are positively related to the brand goodwill.

It can be seen that under the collaborative marketing business model, improving brand goodwill will not only increase the wholesale price and retail price of the products, but also stimulate collaborative marketing and improve the level of effort in big-data marketing services. On the one hand, the increase in double markup will increase the double marginal effect of the supply chain, but on the other hand, consumers can get better big-data marketing services and get a better consumer experience.

**Corollary 3.** Under the collaborative marketing business model, the steady states of *M*'s $w^S(t)$, *RI*'s $p^S(t)$

and $B^S(t)$ are: $w_\infty^S = \frac{(\Phi - \beta K)G_\infty^S}{2\beta}, p_\infty^S = \frac{(3\Phi - \beta K)G_\infty^S}{4\beta}, B_\infty^S = \frac{1}{2\gamma}\left[ r + 2\delta - \left( (r+2\delta)^2 - \frac{\gamma^2(\Phi+\beta K)^2}{2\beta k} \right)^{1/2} \right]G_\infty^S.$

The steady states of *M* and *I*'s optimal value functions are

$$V_{M\infty}^S = \left( (r+2\delta)^2 - \frac{\gamma^2(\Phi+\beta K)^2}{2\beta k} \right)^{-1/2}\frac{(\Phi+\beta K)^2}{8\beta}\left( G_\infty^S \right)^2,$$

$$V_{RI\infty}^S = \frac{k}{4\gamma^2}\left[ r + 2\delta - \left( (r+2\delta)^2 - \frac{\gamma^2(\Phi+\beta K)^2}{2\beta k} \right)^{1/2} \right]\left( G_\infty^S \right)^2.$$

where the steady state of brand goodwill is $G_\infty^S = 0$.

Comparing the situation that the brand goodwill larger than zero under the independent operation scenario and collaborative production mode, the steady state of brand goodwill under the cooperative marketing business model is zero, but it does not mean that this business model is meaningless. Because, combined with Proposition 3, it can be known that the brand goodwill is only present at the moment $t \to +\infty$, and during the normal business operation period $t \in [0, +\infty)$, the brand goodwill is always above zero, and its attenuation index is lower than the other two cases.

An important revelation for supply chain companies is that, on the one hand, collaborative marketing alliance *RI* should look for the sustainable development of big-data marketing to enhance brand goodwill, which is also the key to sustainable profitability. On the other hand, *M* not only obtains the economic benefits of the waste remanufacturing project, but also gets the demand increase brought by the big-data marketing service. However he did not subsidize or share the costs for the

big-data marketing effort invested by *RI*, which can be seen as a free-riding behavior. Undoubtedly, it is easy for the collaborative marketing alliance to lose profitability due to the high cost of big-data marketing and undermine the sustainable development of the supply chain if *M* continues this way. To this end, *M* should be aware of the importance of big-data marketing and enter into a contract with the *RI* to share the cost of their big-data marketing effort to ensure the long-term development of the supply chain.

## 4. Comparative Analysis

This section is based on the equilibrium results of the three business models of independent operation, *N*, collaborative production, *P*, and collaborative marketing, *S* obtained in the forth section. The comparison of steady state of brand goodwill, the optimal wholesale price, retail price and big-data marketing effort as well as profits of supply chain under three business models will be given analytically. And the sensitivity analysis of the optimal decision-making under the three business models w.r.t. the key exogenous parameters will also be revealed to obtain the corresponding enlightenment of enterprise management.

**Proposition 4.** Under the three business models, the relationship between the steady states of brand goodwill is $G_\infty^P > G_\infty^N > G_\infty^S$.

**Proposition 5.** Under the three business models, the relationship between the steady states of the optimal strategies are $w_\infty^N > w_\infty^S$, $B_\infty^P > B_\infty^N > B_\infty^S$.

Propositions 4 and 5 indicate that the brand goodwill has the highest steady state under the collaborative marketing business model *S*, the middle of the independent business model, and the lowest under the collaborative production model. And from Proposition 1-3, the wholesale price is directly proportional to the brand goodwill, so the wholesale price under the collaborative marketing is lower than the independent business situation. According to the analysis, when the supply chain enterprises operate independently, *M*, as the leader of the three enterprises, has first-move advantage over *R* and *I* when they form a collaborative marketing strategic alliance, and can set higher wholesale prices. The steady state of big-data marketing effort under the three business models also has a corresponding size relationship with brand goodwill. It is worth noting that the retail price of products under the independent operation mode and the collaborative production mode is related to various factors, and needs to be analyzed in detail, but the steady-state retail price in joint marketing is always the lowest.

**Proposition 6.** Under the three business models, the relationship between the steady states of the SC's optimal value functions are $\left(V_{MR\infty}^P + V_{I\infty}^P\right) > \left(V_{M\infty}^N + V_{R\infty}^N + V_{I\infty}^N\right) > \left(V_{M\infty}^P + V_{RI\infty}^P\right)$.

It is shown that the total profits of the supply chain in the collaborative production mode in the steady state situation is the highest, the independent business model is second, and the collaborative marketing model is the lowest. When *M* and *R* form a production alliance, the double marginal effect of the supply chain is reduced, and the wholesale cost is saved. At the same time, the Proposition 4-5 can be seen that the steady state of goodwill in the coordinated production scenario is the highest among the three business models, and the goodwill is the key factors affecting consumer demand; on the one hand, directly improving consumer demand, on the other hand, by increasing the return rate of used products, the new demand is generated. Therefore, the collaborative production and operation model greatly stimulates demand and improves profits of supply chain.

**Proposition 7.** The sensitivity analysis of brand goodwill and the optimal decision-making w.r.t. are some key parameters under the three business models shown in Table 2.

**Table 2.** Sensitivity Analysis of Key Parameters.

|  | $\gamma$ | $G_0$ | $\rho$ | $\mu$ | $\varepsilon$ | $\theta$ | $\Delta$ | $\xi$ | $k$ |
|---|---|---|---|---|---|---|---|---|---|
| $G^N$ | ↗ | ↗ | ↗ | ↗ | ↗ | ↗ | ↗ | ↗ | ↘ |
| $G^P$ | ↗ | ↗ | ↗ | ↗ | ↗ | ↗ | ↗ | ↗ | ↘ |
| $G^S$ | ↗ | ↗ | ↗ | ↗ | ↗ | ↗ | ↗ | — | ↘ |
| $w^N$ | ↗ | ↗ | * | ↗ | ↗ | ↗ | ↘ | * | ↘ |
| $w^S$ | ↗ | ↗ | * | ↗ | ↗ | ↗ | * | ↗ | ↘ |
| $p^N$ | ↗ | ↗ | ↗ | ↗ | ↗ | ↗ | * | ↗ | ↘ |
| $p^P$ | ↗ | ↗ | ↗ | ↗ | ↗ | ↗ | * | ↗ | ↘ |
| $p^S$ | ↗ | ↗ | * | ↗ | ↗ | ↗ | * | — | ↘ |
| $B^N$ | ↗ | — | ↗ | ↗ | ↗ | ↗ | ↗ | ↗ | ↘ |
| $B^P$ | ↗ | — | ↗ | ↗ | ↗ | ↗ | ↗ | ↗ | ↘ |

* Note: ↗ represents positive correlation, ↘ represents negative correlation, — represents irrelevant, * represents depending on the situation.

Detailed analysis found that the greater the efficiency of big-data marketing efforts on brand goodwill, the higher brand goodwill, wholesale price, retail price and big-data marketing effort will be, accordingly. The initial brand goodwill has no impact on big-data marketing efforts but only affect the subsequent development of the brand and product development. The higher the initial brand goodwill, the higher the brand goodwill, and the higher the corresponding product price. The higher the correlation coefficient between return rate and goodwill, the higher the brand goodwill and big-data marketing efforts under the business model will be.

The wholesale price of a product depends on the incentives to demand by the recovery. When the return rate of the used products is high, the increase of $\rho$ will increase the wholesale price and increase the marginal benefit of the manufacturer. The retail price is positively correlated with $\rho$ in both the independent operation and the collaborative production mode.

Brand goodwill and the optimal decision of each enterprise are always positively correlated with the reference coefficient of reference price and brand goodwill, $\mu$, the coefficient of influence of return rate on demand, $\varepsilon$, and the direct influence factor of goodwill on demand, $\theta$, and negative related to the cost coefficient of big-data marketing, $k$.

As for the recycling process, higher residual value of used products can always improve brand goodwill and encourage big-data marketing effort of $I$, and the impact on product price depends on different business models and external environment.

The marginal revenue of $I$ always enhances the brand goodwill under the independent operation and collaborative production scenarios in time, but has no influence on the brand goodwill of the cooperative marketing. The marginal revenue of $I$ comes from $R$, so the increase in $\xi$ will also enhance the retail price of the product.

The enlightenment of this business operation is as follows: In the era of data-driven marketing, making good use of big-data marketing technology and improving the marketing efficiency of big-data is one of the most important means to enhance brand goodwill. Specifically, companies strengthen the data analysis capabilities of marketers, consistently use big-data to understand, predict, shape and enhance the customer experience, enhance their perception of the brand, and shape and differentiate the services that big-data marketing brings to consumers and enhance brand goodwill. The size of the initial brand goodwill does not restrict enterprises from using big-data marketing to sell products. Under the massive data, the products are faced with the same big-data marketing tools, choose the right business model, and rationally use the big-data marketing advantage which is the way for companies to win.

## 5. Numerical Analysis

Due to the complexity of the model, some analytical properties are difficult to analyze. This section uses numerical examples to further compare the performance of firms under different business models, and analyzes the impact of price and recycling-related parameters on firms' decision-making and profits, and further analyzes the consumers' surplus and total social welfare under different business models in order to obtain corresponding management insight. To this end, taking into account the model assumptions and generality, set the parameters as follows:

$\gamma = 2, \delta = 0.7, \rho = 0.01, \mu = 1, \beta = 0.5, \varepsilon = 0.5, \theta = 0.5, \Delta = 2, c_T = 0.1, c_M = 0.4, \xi = 0.7, k = 2.$

Figures 2–7 shows the time trajectories of supply chain performance under three business models. Figures 8–12 analyzes the impact of reference price effect factors on firm performance. Figures 13–15 shows the profits' sensitivity trajectory of companies in supply chain w.r.t. marginal residual value.

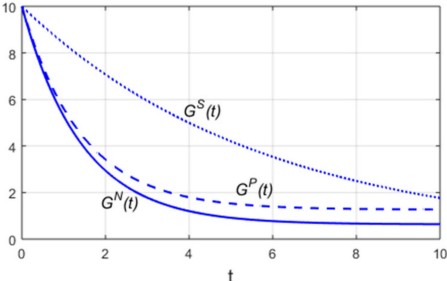

**Figure 2.** Time trajectory of brand goodwill.

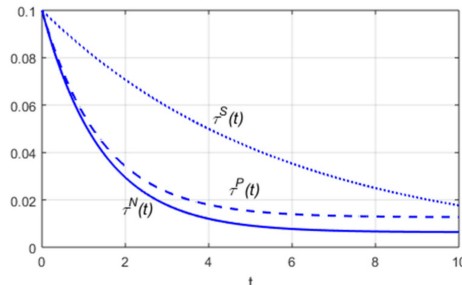

**Figure 3.** Time trajectory of waste product return rate.

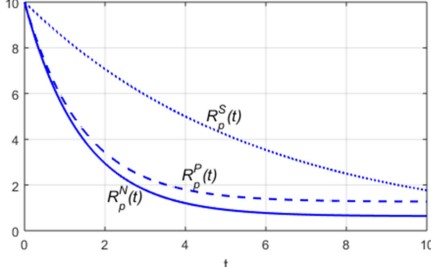

**Figure 4.** Time trajectory of product reference price.

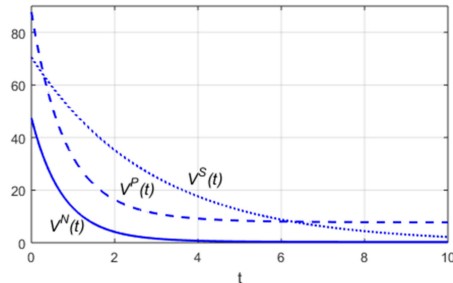

**Figure 5.** Time trajectory of supply chain profit.

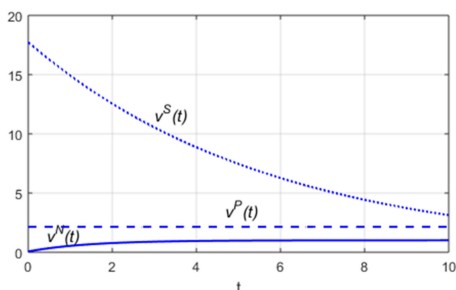

**Figure 6.** Time trajectory of consumer utility.

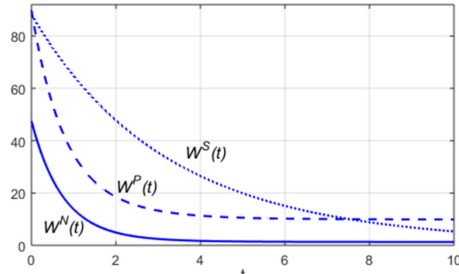

**Figure 7.** Time trajectory of social welfare.

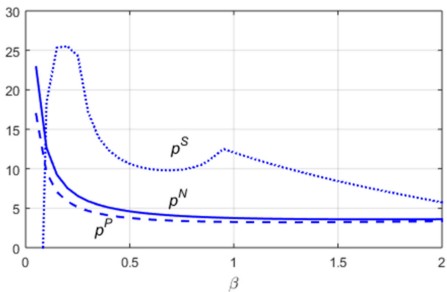

**Figure 8.** R's price with the reference price effect factor, $\beta$.

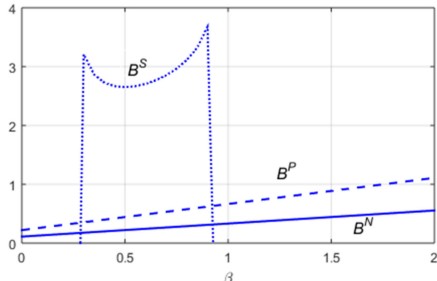

**Figure 9.** I's big-data marketing effort with the reference price effect factor, $\beta$.

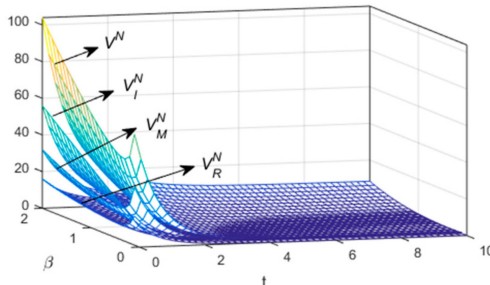

**Figure 10.** Profits under model N with the reference price effect factor, $\beta$ and time, $t$.

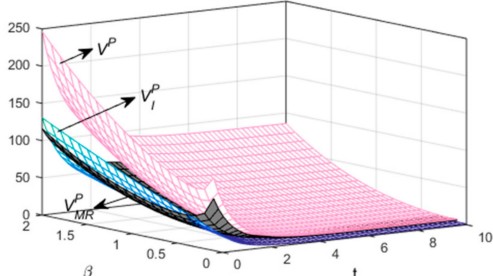

**Figure 11.** Profits under model P with the reference price effect factor, $\beta$ and time, $t$.

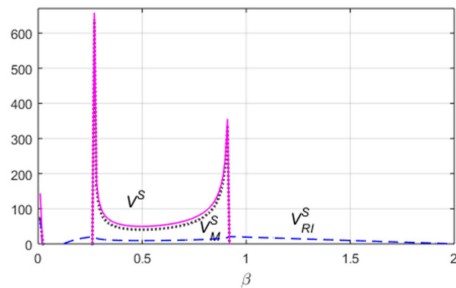

**Figure 12.** Profits under model S with the reference price effect factor, $\beta$.

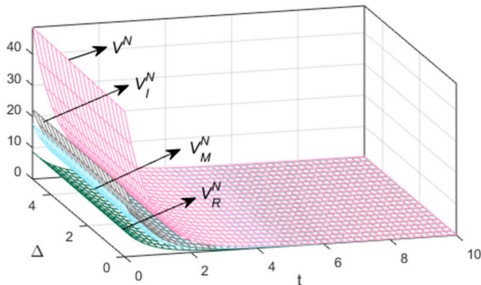

**Figure 13.** Profits under model N with the marginal residual value of used products, $\Delta$ and time, $t$.

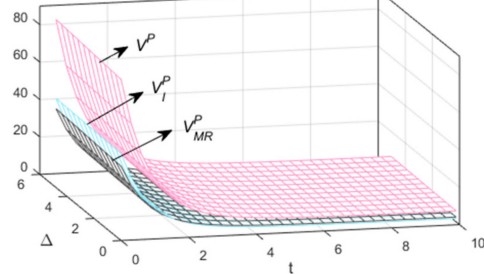

**Figure 14.** Profits under model P with the marginal residual value of used products, $\Delta$ and time, $t$.

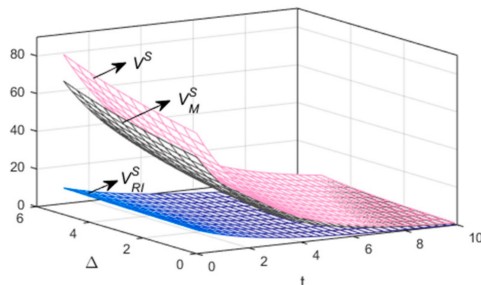

**Figure 15.** Profits under model S with the marginal residual value of used products, Δ and time, *t*.

Figure 2 depicts the time evolution path of brand goodwill under three business models. It can be seen that the brand goodwill of all three business models decays with time, and has the lowest decay rate under the cooperative marketing model while the fastest decay rate in the independent business model. Different from the relationship of goodwill in steady state, the goodwill of the collaborative marketing model is much higher than the other two business models during the normal operation period of the enterprises. The reason is that the cooperative marketing avoids transfer payment $\xi$ between *R* and *I*, collaborative marketing alliance *RI* can allocate more funds to big-data marketing, and big-data marketing is also a key factor to enhance goodwill, so the brand goodwill under the collaborative marketing model is the highest in three. In addition, compared to independent operations mode, *M* and *R* should cooperate with each other in operation which can also take advantage of corporates and enhance brand goodwill.

Since the return rate of used products and the reference price of consumption are positively related to the brand goodwill. On the one hand, consumers will return more used products and enhance environmental benefits under the cooperative marketing model. On the other hand, when *R* and the *I* form a collaborative marketing alliance, it will have all marketing tools of retail price and big-data marketing. Consumers will get more big-data marketing services, better consumer experience, and higher recognition of brands which will produce higher price expectations, as shown in Figures 3 and 4. On the whole, consumers are always expecting higher alliance between companies, which is reflected in higher brand goodwill, higher return rate and much higher willingness to pay.

Figure 5 shows the time trajectories of supply chain profits. Observing that no specific business model has always been the best choice for supply chain enterprises, compared with the independent business model, the alliance business model between enterprises will always improve its economic benefits. The important management inspiration for the company is that choosing the right corporate alliance model at the right time is a strategic choice for the supply chain to have sustainable profitability. In the initial stage, collaborative production is its best choice; while in the growth stage of enterprises, production technology is relatively mature, consumers have a perception of brands, and production links are no longer the main focus of the supply chain. *R* should cooperate with the *I* to co-market products, shape brand image, enhance consumers' willingness to pay for products, and promote more consumers to return used products that have been used in the past, and promote the remanufacturing projects to enterprises and creates the driving force for sustainable development for society, which enables the company to develop steadily. When the supply chain enterprises are in the mature stage, big-data marketing is on the right track, the brand image is stable in the hearts of consumers, and the consumers are diversified and individualized. The demand and collaborative production and operation model will be an important guarantee and the best choice for the supply chain to meet the continuous demand of the consumer market.

Assuming that consumer utility is affected by both the reference quality effect and the big-data marketing service, it can be expressed as $v(t) = \chi_1\left[R_p(t) - p(t)\right] + \chi_2 B(t)$, where $\chi_1$ and $\chi_2$ indicate the influence coefficient of the two respectively. Due to the "addressability" of big-data marketing, Internet platform can conduct targeted and personalized recommendations based on consumers' browsing data and search records in time, so that consumers can find suitable products that suit their needs.

The effectiveness of data marketing for consumers can be several times or even hundred times that of traditional marketing such as pricing. Therefore, set $\chi_1 = 0.2$, $\chi_2 = 5$. The consumer's utility time trajectory is shown in Figure 6. Observing that the collaborative marketing business model has the highest level of big-data marketing effort, it can bring the highest experience utility to consumers, much higher than the collaborative production mode and independent business model.

The sum of supply chain profit (Figure 5) and consumer utility (Figure 6) is the total social welfare, $W(t)$, as shown in Figure 7. It can be seen that whenever an alliance is formed among enterprises, the total social welfare will always be improved. In the initial stage of growth and growth, the society always tends to choose the supply chain of collaborative marketing mode to improve the total social welfare. At the maturity stage, the society has a deep impression on the brand goodwill, so the social welfare is the highest in the collaborative production scenario.

Analysis of the impact of the price effect factor $\beta$ on the optimal retail price of the enterprise and the big-data marketing efforts strategy (Figures 8 and 9) can be found that when consumers' reference price effect is small, $\beta \in [0, 0.2]$, the retail price under the independent operation and collaborative production mode decreases with $\beta$, while the retail price under the collaborative marketing model increases with $\beta$. At this time, the retail price is the highest under the independent business model while the lowest under the collaborative marketing business model. When the reference price effect factor $\beta$ exceeds 0.2, the retail price under the collaborative marketing model will be much higher than the other two situations, and when $\beta \in [0.7, 1.0]$, the retail price will be in a state of increasing first and then decreasing. And when $\beta \geq 1.0$, it gradually falls. As for other scenarios, the retail price has been in a declining state, and lower in collaborative production mode has than the independent operation mode.

The big-data marketing effort under the collaborative production and independent business models always improve with the increase of the reference price effect factor $\beta$. Under the collaborative marketing business model, only when $\beta \in [0.27, 0.92]$, the collaborative marketing alliance will make big-data marketing effort. It can be observed that the reference price effect has the greatest impact on corporate decision-making under the collaborative marketing production model, which is reflected in the fluctuation of retail price and big-data marketing effort. The enlightenment to the enterprise is that when the consumers ' reference price effect is very low, choosing the collaborative marketing business model can effectively alleviate the double marginal effect of the supply chain and stimulate the investment of big-data marketing effort. When the consumption reference price effect is moderate, selecting the collaborative marketing model can maximize the big-data marketing effort. And when the consumer reference quality effect is high, the collaborative production model is the optimal choice for the supply chain.

The sensitivity of each enterprise's profit under the three business models w.r.t. the reference price effect factor $\beta$ is shown in Figures 10–12. Under the independent business model, the profit of *I* always enhances with the increase of the consumer reference price effect. And when $\beta$ is not small, that is $\beta > 0.2$, its profit will be higher than the profit in the other two business models. The profits of *M* and *R* decreases with increasing in $\beta$ when $\beta \in [0, 0.2]$ and increase with $\beta$ when $\beta > 0.2$, meanwhile, *M* gets more economic benefits from recycling and remanufacturing in that situation, and his profit is always higher than *R* 's. Supply chain profit trends are the same as *M* and *R*. It can be seen that under the independent business model, consumers' higher reference price effects are beneficial to enterprises to gain more economic advantages.

Under the cooperative production scenario, the profit of *I* is always positively correlated with $\beta$, and exceeds the profit of collaborative production in $\beta > 0.5$. The profit of the collaborative production alliance is negatively correlated with $\beta$ in $\beta \in [0, 0.5]$, and when $\beta > 0.5$, it is positively related to $\beta$. Profit trend of supply chain is similar to collaborative production alliance's. Under the collaborative marketing business model, the profit of the *RI* is less affected by the consumer reference price effect, while the profit of the *M* and the supply chain fluctuates greatly with $\beta$, only if $\beta \in [0.27, 0.92]$, the two will make a profit. One interval coincides with the sensitivity of big-data marketing efforts on $\beta$. It can

be seen that under the collaborative marketing business model, big-data marketing effort is the key to enhance profits of collaborative alliances and CLSC.

Figures 12–14 shows the relationship between corporates' profits and residual value $\Delta$ in the three types of business models. The analysis shows that regardless of the business model, the profit of the enterprise and the supply chain are positively related to the residual value of the used products, and it is particularly prominent in the collaborative marketing model. The enlightenment to business management is that promoting the recycling of used products to promote remanufacturing projects not only contributes to the improvement of environmental protection benefits, but also the key to achieving greater competitiveness for the company itself.

## 6. Conclusions

As an independent decision-making body, the Internet service platform has enriched the marketing channels of the closed-loop supply chain and diversified the cooperative business model among enterprises. Meanwhile, big-data marketing both reduces the cost of acquiring new customers and retaining old customers. It also saves the labor cost of traditional marketing methods and promotes the rational allocation and effective use of resources. In addition, big-data marketing improves the accuracy of understanding and predicting consumer demand, while avoiding unnecessary cost of inventory. On the other hand, due to its personalized recommendation based on consumer information, the consumer's experience is improved, which also enhances the brand's goodwill and consumers' willingness to return used products. Thus, the return rate of used products increases accordingly, which reduces the input of new materials in the product manufacturing process, and avoids environmental pollution caused by improper landfilling of waste products, whether from the perspective of resource utilization or environmental protection, it all adds to the sustainability of CLSC.

In this context, this paper considers the reference price effect of consumers, and constructs a differential game model under the three modes of independent operation, collaborative production and collaborative marketing between manufacturer, retailer and Internet service platform. By dynamic programming theory, this work obtains the feedback optimal strategies of price and big-data marketing effort, brand goodwill and firms' profits under the three business models. After comparing the three scenarios and the sensitivity analysis of the enterprises' optimal decision-making on key exogenous parameters, three important conclusions can be drawn from this paper:

(1) The retail price of products, the return rate of used products and the profits of the firms under the three business models are all related to the brand goodwill. It can be seen that regardless of the business model, improving the brand goodwill is the key way for each company to profit. To this end, the management insight for enterprises is that every member in CLSC should be committed to improving brand goodwill. As for manufacturer, the use of recyclable environmentally friendly materials from the production source can prolong the product life cycle and reduce the environmental pollution caused by the disposal of waste products. On the other hand, the available parts of waste products returned from consumers have increased. The purpose of these is to increase the residual value of waste products, which has proven to be helpful in improving goodwill. Another effective way for manufacturers is to innovate in production technology, reduce the cost of remanufacturing waste products, and reduce production costs. As for retailers, enhance the correlation between consumer reference prices and brand goodwill, and grasp the formulation of retail prices to make them more consistent with brand positioning, so as to meet consumer expectations of product prices and enhance their loyalty to brands. In addition, to the means of big-data marketing technology, as a member of the supply chain directly facing consumers, enhancing the consumer's experience on the product and improving the ability to serve consumers can also effectively enhance brand goodwill. Not to mention the Internet service platform, it should continue technological innovation, adjust organizational structure, reduce big-data marketing costs, and improve of big-data marketing accuracy, which can accurately serve consumers, help them find products suitable for them, and increase consumer brand viscosity.

In addition to the efforts of various enterprises, cooperation between enterprises also makes an important contribution to the construction of brand goodwill. Collaborative production and collaborative marketing business models can stimulate the big-data marketing efforts of the Internet service platform to enhance the consumer experience and thus enhance brand goodwill.

(2) Under the background of big-data marketing, consumers reference price effect has an important impact on enterprises' strategies and economic performance. In terms of strategy formulation, when the reference price effect is small, retailers will reduce retail prices and mitigate the double marginal effect of the supply chain under the collaborative production and collaborative marketing business model; the big-data marketing effort under the independent operation and collaborative production mode will follow up with the increase of the reference price effect and the big-data marketing effort will exist only when the reference price effect factor is within a certain range.

(3) Under the influence of big-data marketing and consumer reference price effect, the optimal business model selection of enterprises should be determined according to the situation. When enterprises are in the initial stage, it should choose the collaborative production mode to help it save operating costs and enter the market. When enterprises are in the growth stage, the retailer should cooperate with the Internet service platform to conduct big-data marketing to shape the brand image and increase the willingness of consumers to pay. When enterprises are in maturity, the brand image is stable, and the collaborative production mode can meet the diversified needs of consumers.

In summary, whether it is to include Internet service platform into CLSC, use its big-data marketing to accurately target consumer needs, or fully consider the consumer's reference price effect, make marketing truly consumer-oriented, and promote vertical cooperation among enterprises, all of these are to improve the sustainability of CLSC. It is embodied in the following aspects: **Economic sustainability**. Taking into account the consumer's reference price effect, as well as the diversity and personalization of consumer demand, adopting big-data marketing can transform corporate marketing activities into consumer-oriented scientific decisions. Through dynamic cooperation between enterprises, production of marketable products, recommending products timely and appropriately that meet the needs to consumers through multiple channels, realize precise conversion of sales efficiently, enhance market competitiveness, and win the long-term benefits of CLSC. **Environmental sustainability**. Adopting big-data marketing technology can effectively predict and accurately locate consumer needs, avoid unnecessary product production, and save resources. On the other hand, big-data marketing gives companies a broader way to promote the importance of remanufacturing engineering to consumers, which will help more consumers return used products, increase their utilization rate, avoid the input of new materials, and avoid the environmental pollution caused by the landfill of used products. Additionally, through cooperation between enterprises, big-data marketing and respective functions of each CLSC members can be maximized. **Social sustainability**. The application of big-data marketing technology, and in-depth cooperation between companies can help manufacturers fulfill more social responsibilities, and affected by products' brand goodwill, consumers can feel the greenness of waste product recycling, have a stronger sense of social responsibility to participate in the product recycling process, which has a profound impact on creating a green, environmentally responsible society. Thus, it can achieve the triple-bottom line of CLSC.

**Author Contributions:** Formal analysis, D.M.; methodology, D.M.; supervision, J.H.; writing—original draft, D.M.; writing—review and editing, J.H. All authors have read and agreed to the published version of the manuscript.

**Funding:** This research was funded by the National Natural Science Foundation of China with Grant No. 71771129.

**Conflicts of Interest:** The authors declare no conflict of interest.

## Appendix A

**Proof of Proposition 1:** In order to obtain the optimal business strategies of enterprises, the strategy of the *I* should be solved first by means of the inverse induction method. According to the Berman's continuous dynamic programming theory [43], for any state $G \geq 0$, there is a continuous differentiable value function, satisfying Hamilton-Jacobi-Bellman (hereinafter abbreviated as HJB) equation

$$rV_I^N = \max_{B^N}\left\{\xi\left[\Phi G - \beta p^N\right] - \frac{k\left(B^N\right)^2}{2} + \frac{\partial V_I^N}{\partial G}\left[\gamma B^N - \delta G\right]\right\} \tag{6}$$

For the sake of simplicity of the symbol, *t* will be omitted from the expression below. Where $V_I^N$ is the optimal value function of the Internet service platform, indicating its total profit during the planning period, $\frac{\partial V_I^N}{\partial G}$ is the first derivative of its optimal value function with respect to the state variable *G*, and expresses the margin contribution of the unit change of the brand goodwill to its profit. It can be seen that under the dynamic decision-making environment, enterprises not only consider the immediate benefits, but also comprehensively consider the impact of future changes in brand goodwill on long-term profits, and formulate their decisions with the long-term profit as the goal. Using first-order optimality conditions, we can obtain

$$B^N = \frac{\gamma}{k}\frac{\partial V_I^N}{\partial G} \tag{7}$$

Next, substituting Equation (7) into *R*'s target functional to construct his HJB equation:

$$rV_R^N = \max_{p^N}\left\{\left[p^N - w^N - \xi\right]\left[\Phi G - \beta p^N\right] + \frac{\partial V_R^N}{\partial G}\left[\gamma B^N - \delta G\right]\right\} \tag{8}$$

*R* 's retail price response function from first-order optimality conditions is:

$$p^N = \frac{\Phi G + \beta\left(w^N + \xi\right)}{2\beta} \tag{9}$$

Substituting Equations (7) and (9) into the *M*'s target functional, *M*'s HJB equation is:

$$rV_M^N = \max_{w^N}\left\{\left[w^N + KG\right]\left[\frac{\Phi G - \beta\left(w^N + \xi\right)}{2}\right] + \frac{\partial V_M^N}{\partial G}\left[\gamma B^N - \delta G\right]\right\} \tag{10}$$

*M* 's optimal wholesale price strategy is obtained by the first-order optimal condition of Equation (10):

$$w^N = \frac{(\Phi - \beta K)G - \beta\xi}{2\beta} \tag{11}$$

Substituting Equation (11) into Equation (9), 's optimal retail price strategy is

$$p^N = \frac{(3\Phi - \beta K)G + \beta\xi}{4\beta} \tag{12}$$

Substituting Equations (7), (11), and (12) into Equations (10), (8), and (6), respectively, can obtain equations that the optimal function of $M$, $R$, and $I$ satisfies.

$$
\begin{aligned}
rV_M^N &= \left[\frac{(\Phi+\beta K)G-\beta\xi}{2\beta}\right]\left[\frac{(\Phi+\beta K)G-\beta\xi}{4}\right] + \frac{\partial V_M^N}{\partial G}\left[\frac{\gamma^2}{k}\frac{\partial V_I^N}{\partial G}-\delta G\right] \\
rV_R^N &= \left[\frac{(\Phi+\beta K)G-\beta\xi}{4\beta}\right]\left[\frac{(\Phi+\beta K)G-\beta\xi}{4}\right] + \frac{\partial V_R^N}{\partial G}\left[\frac{\gamma^2}{k}\frac{\partial V_I^N}{\partial G}-\delta G\right] \\
rV_I^N &= \xi\left[\frac{(\Phi+\beta K)G-\beta\xi}{4}\right] - \frac{\gamma^2}{2k}\left[\frac{\partial V_I^N}{\partial G}\right]^2 + \frac{\partial V_I^N}{\partial G}\left[\frac{\gamma^2}{k}\frac{\partial V_I^N}{\partial G}-\delta G\right]
\end{aligned}
\tag{13}
$$

According to the structure of Equation (13), the optimal value functions of the three are respectively $V_M^N = f_1 G^2 + f_2 G + f_3$, $V_R^N = g_1 G^2 + g_2 G + g_3$ and $V_I^N = h_1 G + h_2$. Among them $f_1, f_2, f_3, g_1, g_2, g_3$ and $h_1, h_2$ are the constant coefficients of the value function. Substituting the value function and its derivative w.r.t. $G$ for equation (13), and obtaining the undetermined coefficient according to the identity relationship:

$$
\begin{aligned}
&f_1 = \frac{(\Phi+\beta K)^2}{8(r+2\delta)\beta}, f_2 = \frac{\xi\gamma^2(\Phi+\beta K)^3}{16(r+\delta)^2(r+2\delta)\beta k} - \frac{\xi(\Phi+\beta K)}{4(r+\delta)}, \\
&f_3 = \frac{\beta\xi^2}{8r} + \frac{\gamma^2\xi(\Phi+\beta K)}{4r(r+\delta)k}\left[\frac{\xi\gamma^2(\Phi+\beta K)^3}{16(r+\delta)^2(r+2\delta)\beta k} - \frac{\xi(\Phi+\beta K)}{4(r+\delta)}\right], \\
&g_1 = \frac{(\Phi+\beta K)^2}{16(r+2\delta)\beta}, g_2 = \frac{\xi\gamma^2(\Phi+\beta K)^3}{32(r+\delta)^2(r+2\delta)\beta k} - \frac{\xi(\Phi+\beta K)}{8(r+\delta)}, \\
&g_3 = \frac{\beta\xi^2}{16r} + \frac{\gamma^2\xi(\Phi+\beta K)}{4r(r+\delta)k}\left[\frac{\xi\gamma^2(\Phi+\beta K)^3}{32(r+\delta)^2(r+2\delta)\beta k} - \frac{\xi(\Phi+\beta K)}{8(r+\delta)}\right], \\
&h_1 = \frac{\xi(\Phi+\beta K)}{4(r+\delta)}, h_2 = \frac{\gamma^2\xi^2(\Phi+\beta K)^2}{32r(r+\delta)^2 k} - \frac{\beta\xi^2}{4r}.
\end{aligned}
\tag{14}
$$

The optimal big-data marketing effort for $I$ is:

$$
B^N = \frac{\xi\gamma(\Phi+\beta K)}{4(r+\delta)k}
\tag{15}
$$

Substituting Equation (15) into the equation of state

$$
\dot{G}^N(t) = \frac{\xi\gamma^2(\Phi+\beta K)}{4(r+\delta)k} - \delta G^N(t), G(0) = G_0 > 0
\tag{16}
$$

Solving the differential equation can obtain the optimal goodwill time evolution trajectory in independent business mode:

$$
G^N(t) = e^{-\delta t}\left[G_0 - \frac{\xi\gamma^2(\Phi+\beta K)}{4\delta(r+\delta)k}\right] + \frac{\xi\gamma^2(\Phi+\beta K)}{4\delta(r+\delta)k}
\tag{17}
$$

Substituting equation (17) into optimal strategy (11) and (12) can obtain the optimal decision of the enterprise. At the same time, with the undetermined coefficient (14), the profits of $M$, $R$ and $I$ can be obtained. In addition, it should be noted that in order to ensure $M$'s normal wholesale activities, it is necessary to ensure $w^N > 0$, that is $(\Phi-\beta K)G^N > \beta\xi > 0$, and as $G^N > 0$, so $\Phi > \beta K$.□

**Proof of Proposition 4** From the Corollary 1–3, we can easily get that $G_\infty^P - G_\infty^N = \frac{\xi\gamma^2(\Phi+\beta K)}{4\delta(r+\delta)k} > 0 = G_\infty^S$.□

**Proof of Proposition 5** Also from the Corollary 1–3, we take the subtraction between the steady states of optimal strategies under different business models, the results are: $w_\infty^N - w_\infty^S = \frac{\xi\gamma^2(\Phi+\beta K)(\Phi-\beta K)}{8\delta(r+\delta)\beta k} - \frac{\xi}{2} > 0$, $B_\infty^P - B_\infty^N = \frac{\xi\gamma(\Phi+\beta K)}{4(r+\delta)k} > 0 = B_\infty^S$. When $\Phi - 3\beta K < -\frac{4\delta(r+\delta)\beta\xi k}{\xi\gamma^2(\Phi+\beta K)}$, $p_\infty^N - p_\infty^P = \frac{-\xi\gamma^2(\Phi+\beta K)(\Phi-3\beta K)-4\delta(r+\delta)\beta\xi k}{4\beta} > 0$; or otherwise, $p_\infty^N \le p_\infty^P$, $p_\infty^N > p_\infty^S = 0$, $p_\infty^P > p_\infty^S = 0$
□

**Proof of Proposition 6 :** It is clear to see that from Corollary 1–3:

$$
\begin{aligned}
V_{MR\infty}^{P} + V_{I\infty}^{P} - \left(V_{M\infty}^{N} + V_{R\infty}^{N} + V_{I\infty}^{N}\right) = {}& \frac{13(\Phi+\beta K)^2}{64(r+2\delta)\beta}\left(\frac{\xi\gamma^2(\Phi+\beta K)}{2\delta(r+\delta)k}\right)^2 \\
& + \left[\frac{13\gamma^2\xi(\Phi+\beta K)^3}{64(r+\delta)^2(r+2\delta)\beta k} - \frac{5\xi(\Phi+\beta K)}{16(r+\delta)}\right]\frac{\xi\gamma^2(\Phi+\beta K)}{2\delta(r+\delta)k} + \frac{13\beta\xi^2}{16r} \\
& + \frac{\gamma^2\xi(\Phi+\beta K)}{2r(r+\delta)k}\left[\frac{13\gamma^2\xi(\Phi+\beta K)^3}{64(r+\delta)^2(r+2\delta)\beta k} - \frac{5\xi(\Phi+\beta K)}{16(r+\delta)}\right] \\
+ \frac{3\xi(\Phi+\beta K)}{8(r+\delta)}\frac{\xi\gamma^2(\Phi+\beta K)}{2\delta(r+\delta)k} {}& + \frac{31\xi^2\gamma^2(\Phi+\beta K)^2}{32r(r+\delta)^2 k} - \frac{\beta\xi^2}{4r} > 0 = \left(V_{M\infty}^{P} + V_{RI\infty}^{P}\right)
\end{aligned}
$$

.□

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
