# Peer review of "Research on Collaborative Management Strategies of Closed-Loop Supply Chain under the Influence of Big-Data Marketing and Reference Price Effect"

_sustainability, doi:10.3390/su12041685_

Round 1

Reviewer 1 Report

This is an interesting topic and approach to the research study. Enhancing the introduction, defining key terms, and linking the conclusion to the research questions and the focus of the journal will greatly improve the research.  Below are recommendations for revision:

P1 - Define closed loop supply chain.  There are a number of terms such as closed loop supply chain and goodwill that are used throughout the paper and need to be clearly defined (with a citation) early in the paper

P1 – Management research – why is this there? Is it a heading?  Why is there a cite? 

P2 – Remove specific names when referring to professionals (for example, Sean and Jennifer).  Unless you conducted interviews with these individuals, use their title or a reference to the source material. 

P2 - Consider other citations for the impact of big data. McKinsey and Harvard Business Review have multiple articles.

P2 – What is JD? 

P2 – saling process?  Is this supposed to be sales process? 

P 3 – 4 – Summarize this section.  No need to detail every existing study.  Just say existing studies have found (with citations) and this is why this is different. This should be a brief paragraph, not several pages.

P4 – Clarify contributions.  Grammar and numbering of contributions are not clear.  The customer-centric contribution is too general and it is not clear how it relates to the paper. 

P4 – The research questions and results should be stated prior to the contribution.

P 8 – Grammar. But the it, line 282

Conclusion -  Provide more detail on how to improve brand goodwill. This is a key component of the findings, provide more specifics on what companies can do to enhance goodwill 

Conclusion - Tie to back to sustainability and the importance of the close loop supply chain. The focus of the journal is sustainability and it is barely references throughout the paper and the conclusion.  

Author Response

Dear professor:

Thank you very much for your valuable suggestions for this article, and we will modify the manuscript “Research on Collaborative Management Strategies of Closed-loop Supply Chain under the Influence of Big-Data Marketing and Reference Price Effect”(700084) according to your comments and make the following feedback:

P1 - Define closed loop supply chain.  There are a number of terms such as closed loop supply chain and goodwill that are used throughout the paper and need to be clearly defined (with a citation) early in the paper

Response: Closed-loop supply chain is defined as” Closed-loop supply chain(CLSC), as a branch of the supply chain(SC), integrates forward selling process in classical SC , backward activities(e.g., product acquisition, reverse logistics, points of use and disposal, testing, sorting, refurbishing, recovery, recycling) as well as re-marketing and re-selling into a unique system[1,2,3]. There both economic and non-economic motivation for focal firms to implement CLSC4].” in first paragraph in introduction. And brand goodwill is defined as” It refers to the comprehensive image of a brand in the minds of consumers-including its attributes, quality, grade (taste), culture, personality, etc., and can be considered a significant asset of a firm.[17]” in line 59.

1.Guide,V.D.R.Jr. Production planning and control for remanufacturing: industry practice and research needs. J. Oper. Manag.,2000, 18,467–483. [CrossRef]

Wassenhove, L.N.V.;Guide, Jr.V.D.R. The evolution of closed-loop supply chain research. Oper. Res, 2009, 57,10-18.[CrossRef]

3.Souza, G.C. Closed-loop supply chains: a critical review, and future research. Decis. Sci. 2013, 44,7–38[CrossRef]

4.De Giovanni, P., & Zaccour, G. A selective survey of game-theoretic models of closed-loop supply chains. 4OR, 2019, 1-44. [CrossRef]

Johnson, L. T.; Petrone, K. R. Commentary: is goodwill an asset? Acc.Hor. 1998, 12..

P1 – Management research – why is this there? Is it a heading?  Why is there a cite? 

Response: Here is a writing error, which has been adjusted accordingly. And the cite is for the last sentence, see details in line 45-48.

P2 – Remove specific names when referring to professionals (for example, Sean and Jennifer).  Unless you conducted interviews with these individuals, use their title or a reference to the source material. 

Response: I am sorry for this irregular reference, and have adjusted accordingly according to professor’s requirements:” It is pointed out that the effective use of big data can help companies make better decisions [13,14], data analysis companies and third-party Internet service platforms came into being, which also revolutionized closed-loop supply chain management.” And” Some research even defines marketing as consumer-oriented science [12,20], that is, only by truly understanding the hearts and needs of consumers can enterprises make more realistic and competitive decisions. “

12.Smart Cities: Issues and Challenges: Mapping Political, Social and Economic Risks and Threats. Elsevier, 2019.

13.Adele S. The Analytical Marketer: How to Transform Your Marketing Organization. Harv. Bus. Revi.Press.2016. [CrossRef]

14.Chong, A. Y. L.; Ch’ng, E.; Liu, M. J.; et al. Predicting consumer product demands via Big Data: the roles of online promotional marketing and online reviews. Intern. J. Prod. Res. 2017, 55, 5142-5156.

20.Volkov, M. Successful relationship marketing: Understanding the importance of complaints in a consumer-oriented paradigm. Probl.Persp. Manag. 2004: 113-123.

P2 - Consider other citations for the impact of big data. McKinsey and Harvard Business Review have multiple articles.

Response: In order to fully prove the impact of big data and the application of big-data marketing in the practice world, literatures 10-12,14,16 are supplemented accordingly, and the expression has also been adjusted accordingly:” In recent years, new digital marketing channels based on e-commerce and social media have emerged, data has exploded, and more and more companies are beginning to recognize the value of big data [9,10,11,12]. It is pointed out that the effective use of big data can help companies make better decisions [13,14], data analysis companies and third-party Internet service platforms came into being, which also revolutionized closed-loop supply chain management. More and more companies choose to cooperate with third-party Internet service platforms in order to obtain higher-quality user information and understand and predict customer needs more accurately [15]. Unilever, a publicly traded company that sells food and laundry products, even created a new marketing executive position, Big-Data Marketing Officer, to drive corporate marketing reform [16].”

9.Liu, P., & Yi, S. P. Pricing policies of green supply chain considering targeted advertising and product green degree in the big-data environment. J. Clean. Prod. 2017, 164, 1614-1622. [CrossRef]

10.Côrte-Real, N.; Oliveira, T.; Ruivo, P. Assessing business value of Big Data Analytics in European firms. J Bus. Res. 2017, 70, 379-390. [CrossRef]

11.Grover, V.; Chiang ,R. H. L.; Liang, T. P., et al. Creating strategic business value from big data analytics: A research framework. J. Manag. Inform. Syst. 2018, 35: 388-423.

12.Smart Cities: Issues and Challenges: Mapping Political, Social and Economic Risks and Threats. Elsevier, 2019.

13.Adele S. The Analytical Marketer: How to Transform Your Marketing Organization. Harv. Bus. Revi.Press.2016. [CrossRef]

14.Chong, A. Y. L.; Ch’ng, E.; Liu, M. J.; et al. Predicting consumer product demands via Big Data: the roles of online promotional marketing and online reviews. Intern. J. Prod. Res. 2017, 55, 5142-5156.

15.Liu Z.H., & Zhang Q.L. Review of Big-data Technology Research. J. Zhejiang Univ. 2014,48, 957-972. [CrossRef]

16.Traditional CMO eliminated, Unilever establishes new "chief digital marketing officer". Mark.Sit. 2020,1,12-13.

P2 – What is JD? 

Response: JD, on behalf of Beijing Jingdong Century Trading Co., Ltd., a Chinese self-employed e-commerce company, has established in-depth cooperation strategies with many well-known brands and used its big-data marketing technology to obtain the company's long-term profitability, so it is referenced here as a successful practice case. In order to facilitate understanding, corresponding notes have been made in the text, see more details in line 64-65.

P2 – saling process?  Is this supposed to be sales process? 

Response: Another writing error, it is supposed to be sales process. Corresponding changes have been made in the article,see line 73.

P 3 – 4 – Summarize this section.  No need to detail every existing study.  Just say existing studies have found (with citations) and this is why this is different. This should be a brief paragraph, not several pages.

Response:According to the opinions of professor, the literature review has been streamlined, and the differences between existing research and this article have been compared to highlight the innovation of this article. For more details, see the blue font in the literature review section (in line121-132, 143-155, 161-163).

P4 – Clarify contributions.  Grammar and numbering of contributions are not clear.  The customer-centric contribution is too general and it is not clear how it relates to the paper. 

Response: Contribution has been adjusted to” The main contributions of this paper are: 1) further enrich the research results of the dynamic CLSC, expand existing research in the influence of ERP in consumers’ shopping process; 2) incorporate Internet service platform as an independent game subject into the CLSC and use its big-data marketing as a new marketing method; 3)and further explore the optimal vertical cooperation between enterprises to achieve economic, environmental and social benefits of CLCS.” The usage of customer-centric in the text is a bit inappropriate. The article originally wanted to emphasize that in order to make more realistic decision-making, companies should start by studying consumer behavior, and reference price effect is common in consumer shopping. Including consideration of it will make corporate decisions more realistic, and it will also bring richer management insight to corporate decisions. This point has been emphasized in the research background.

P4 – The research questions and results should be stated prior to the contribution.

Response:The research questions and results has been moved prior to the contribution. See line 86-101.

P 8 – Grammar. But the it, line 282

Response:”But the it” has been modified as “but it is…” , now in line 290.

Conclusion -  Provide more detail on how to improve brand goodwill. This is a key component of the findings, provide more specifics on what companies can do to enhance goodwill 

Response: More details specifics on what companies can do to enhance goodwill has been add to the conclusion:” To this end, the management insight for enterprises is that every member in CLSC should be committed to improving brand goodwill: As for manufacturer, the use of recyclable environmentally friendly materials from the production source can prolong the product life cycle and reduce the environmental pollution caused by the disposal of waste products. On the other hand, the available parts of waste products returned from consumers have increased. The purpose of these is to increase the residual value of waste products, which has proven to be helpful in improving goodwill. Another effective way for manufacturer is to innovate in production technology, reduce the cost of remanufacturing waste products, and reduce production costs. As for retailer, enhance the correlation between consumer reference prices and brand goodwill, and grasp the formulation of retail prices to make them more consistent with brand positioning, so as to meet consumer expectations of product prices and enhance their loyalty to brands. In addition, in addition to the means of big data marketing technology, as a member of the supply chain directly facing consumers, enhancing the consumer's experience on the product and improving the ability to serve consumers can also effectively enhance brand goodwill. Not to mention the Internet service platform, it should continue technological innovation, adjust organizational structure, reduce big-data marketing costs, and improve of big data marketing accuracy, which can accurately serve consumers, help them find products suitable for them, and increase consumer brand viscosity. In addition to the efforts of various enterprises, cooperation between enterprises also makes an important contribution to the construction of brand goodwill: Collaborative production and collaborative marketing business models can stimulate the big-data marketing efforts of the Internet service platform to enhance the consumer experience and thus enhance brand goodwill.” In line 617-638.

Conclusion - Tie to back to sustainability and the importance of the close loop supply chain. The focus of the journal is sustainability and it is barely references throughout the paper and the conclusion. 

Response: In order to back to sustainability and the importance of CLSC, more details has been add to the conclusion:” In summary, whether it is to include Internet service platform into CLSC, use its big-data marketing to accurately target consumer needs, or fully consider the consumer's reference price effect, make marketing truly consumer-oriented, and promote vertical cooperation among enterprises, all of these are to improve the sustainability of CLSC. It is embodied in the following aspects: Economic sustainability. Taking into account the consumer's reference price effect, as well as the diversity and personalization of consumer demand, adopting big-data marketing can transform corporate marketing activities into consumer-oriented scientific decisions. Through dynamic cooperation between enterprises, production of marketable products, recommending products timely and appropriately that meet the needs to consumers through multiple channels, realize precise conversion of sales efficiently, enhance market competitiveness, and win the long-term benefits of CLSC. Environmental sustainability. Adopting big-data marketing technology can effectively predict and accurately locate consumer needs, avoid unnecessary product production, and save resources. On the other hand, big-data marketing gives companies a broader way to promote the importance of remanufacturing engineering to consumers, which will help more consumers return used products, increase their utilization rate, avoid the input of new materials, and avoid the environmental pollution caused by the landfill of used products. And through cooperation between enterprises, big-data marketing and respective functions of each CLSC members can be maximized. Social sustainability. The application of big-data marketing technology, and in-depth cooperation between companies can help manufacturer fulfill more social responsibilities, and affected by products’ brand goodwill, consumers can feel the greenness of waste product recycling, have a stronger sense of social responsibility to participate in the product recycling process, which has a profound impact on creating a green, environmentally responsible society. Thus it can achieve the triple bottom line of CLSC.” In line 655-677.

Thank you again for your valuable suggestions, and I wish you a smooth work and a happy life.

Yours sincerely,

Ma Deqing

Reviewer 2 Report

The idea to analyze reference prices in a goodwill model is interesting. Unfortunately, I have serious objects to the model and to the text of the paper:

1) explanation of the term "closed-loop supply chain" is insufficient, the selection of literature is superficial,

2) using the concept of "big-data marketing", although it is becoming more and more fashionable, there is no justification in this model. The authors assumed that marketing activities increase goodwill as it is in any model based on the idea of Nerlove and Arrow. The reader can not find any reason why in this model it should be big-data marketing,

3) there is no sufficient reason why the online platform makes money from brand goodwill,

4) the  first and third scenarios are well documented in supply chain literature, however, the authors of this papers did not mention about them; the second scenario is surprising because it is hard to imagine that a retailer has a direct impact on production and a producer on sale,

5) in games with an infinite horizon, time-variable strategies never appear, why do they occur this time?

Therefore, I think that despite choosing an interesting topic, the paper has many shortcomings and is not suitable for publication in this form. I encourage the authors to work hard on this paper. This is a good start.

Author Response

Dear professor:

Thank you very much for your valuable suggestions for this article, and we will modify the manuscript “Research on Collaborative Management Strategies of Closed-loop Supply Chain under the Influence of Big-Data Marketing and Reference Price Effect”(700084) according to your comments and make the following feedback:

1) explanation of the term "closed-loop supply chain" is insufficient, the selection of literature is superficial.

Response: Closed-loop supply chain is defined as” Closed-loop supply chain(CLSC), as a branch of the supply chain(SC), integrates forward selling process in classical SC , backward activities(e.g., product acquisition, reverse logistics, points of use and disposal, testing, sorting, refurbishing, recovery, recycling) as well as re-marketing and re-selling into a unique system[1,2,3]. There both economic and non-economic motivation for focal firms to implement CLSC4].” in first paragraph in introduction. And this article has also adjusted the discussion of closed-loop supply chain application in the introduction(in line 30-42) and summarized the necessity of closed-loop supply chain management:” Because of its triple-bottom line of economic, social and environmental [7], CLSC has shown far more advantages than the traditional SC. Therefore, the management of CLSC has also become a focus in the theoretical and practical circles.”.

1.Guide,V.D.R.Jr. Production planning and control for remanufacturing: industry practice and research needs. J. Oper. Manag.,2000, 18,467–483. [CrossRef]

Wassenhove, L.N.V.;Guide, Jr.V.D.R. The evolution of closed-loop supply chain research. Oper. Res, 2009, 57,10-18.[CrossRef]

3.Souza, G.C. Closed-loop supply chains: a critical review, and future research. Decis. Sci. 2013, 44,7–38[CrossRef]

4.De Giovanni, P., & Zaccour, G. A selective survey of game-theoretic models of closed-loop supply chains. 4OR, 2019, 1-44. [CrossRef]

7.De Giovanni, P.; Reddy, P. V. & Zaccour, G. Incentive strategies for an optimal recovery program in a closed-loop supply chain. Eur. J. Oper. Res. 2016, 249, 605-617. [CrossRef]

2) using the concept of "big-data marketing", although it is becoming more and more fashionable, there is no justification in this model. The authors assumed that marketing activities increase goodwill as it is in any model based on the idea of Nerlove and Arrow. The reader can not find any reason why in this model it should be big-data marketing,

Response: In order to fully prove the impact of big data and the application of big-data marketing in the practice world, literatures 10-12,14,16 are supplemented accordingly, and the expression has also been adjusted accordingly:” In recent years, new digital marketing channels based on e-commerce and social media have emerged, data has exploded, and more and more companies are beginning to recognize the value of big data [9,10,11,12]. It is pointed out that the effective use of big data can help companies make better decisions [13,14], data analysis companies and third-party Internet service platforms came into being, which also revolutionized closed-loop supply chain management. More and more companies choose to cooperate with third-party Internet service platforms in order to obtain higher-quality user information and understand and predict customer needs more accurately [15]. Unilever, a publicly traded company that sells food and laundry products, even created a new marketing executive position, Big-Data Marketing Officer, to drive corporate marketing reform [16].”

As for big-data marketing in the dynamics of brand goodwill based on N-A model, it is worth pointing out that although the N-A model is used here to describe the goodwill dynamics, the marketing method of advertising has been replaced by big-data marketing. The reason is that traditional marketing methods are based on small data. The lag and limitation are obvious, and due to the diversity of consumer demand for products and the timeliness as well as the multi-platformness of their shopping decisions, based on big data analysis, firms can accurately predict consumer demand and implement personalized and precise marketing. The superiority of big-data marketing technology is evident. What’s more, big-data marketing technology requires high expertise and high operating costs. It is difficult for traditional retail companies to develop independently in a short period of time. Generally, they choose to cooperate with third-party Internet service platforms with mature big-data marketing technologies, which also is the basis of this article model. In order to facilitate the reader's understanding, the above reasons are also reflected in the model construction in the article, see more details in line 196-203.

9.Liu, P., & Yi, S. P. Pricing policies of green supply chain considering targeted advertising and product green degree in the big-data environment. J. Clean. Prod. 2017, 164, 1614-1622. [CrossRef]

10.Côrte-Real, N.; Oliveira, T.; Ruivo, P. Assessing business value of Big Data Analytics in European firms. J Bus. Res. 2017, 70, 379-390. [CrossRef]

11.Grover, V.; Chiang ,R. H. L.; Liang, T. P., et al. Creating strategic business value from big data analytics: A research framework. J. Manag. Inform. Syst. 2018, 35: 388-423.

12.Smart Cities: Issues and Challenges: Mapping Political, Social and Economic Risks and Threats. Elsevier, 2019.

13.Adele S. The Analytical Marketer: How to Transform Your Marketing Organization. Harv. Bus. Revi.Press.2016. [CrossRef]

14.Chong, A. Y. L.; Ch’ng, E.; Liu, M. J.; et al. Predicting consumer product demands via Big Data: the roles of online promotional marketing and online reviews. Intern. J. Prod. Res. 2017, 55, 5142-5156.

15.Liu Z.H., & Zhang Q.L. Review of Big-data Technology Research. J. Zhejiang Univ. 2014,48, 957-972. [CrossRef]

16.Traditional CMO eliminated, Unilever establishes new "chief digital marketing officer". Mark.Sit. 2020,1,12-13.

3) there is no sufficient reason why the online platform makes money from brand goodwill.

Response: In the article, will usefor big-data marketing to consumers and charged by with the service fee [9,41], where represents the unit service rate. The marginal benefit of is, and his total benefit is,where. Online platform does not make money directly from brand goodwill, but he can obtain more demand through the promotion of goodwill, thereby increasing revenue. The main way to improve goodwill is his big- data marketing technology.

9.Liu, P., & Yi, S. P. Pricing policies of green supply chain considering targeted advertising and product green degree in the big-data environment. J. Clean. Prod. 2017, 164, 1614-1622. [CrossRef]

41.Xiang, Z. & Xu, M. Dynamic cooperation strategies of the closed-loop supply chain involving the Internet service platform. J. Clean. Prod.2019, 220, 1180-1193. [CrossRef]

4) the  first and third scenarios are well documented in supply chain literature, however, the authors of this papers did not mention about them; the second scenario is surprising because it is hard to imagine that a retailer has a direct impact on production and a producer on sale.

Response: For the first scenario, Independent Business Scenario(Model ), corresponding supporting literature and descriptions have been supplemented as” This business model is in line with our general understanding of business operations and is widely used in the researches [5-8].” And for the third model, corresponding supporting literature and descriptions have been supplemented as” This business model is adopted by some powerful companies under big-data-driven marketing. They generally have sufficient funds and manpower, and also have enough customer data to be able to carry out big-data marketing independently. Such as Unilever independently established a digital marketing department [16] and JD build a precision marketing framework[20], all of which aim to use the new technology of big-data marketing to more accurately put products into the hands of customers and enhance their shopping experience.” As for the second model questioned by professor, the reason for this joint production mode is that with the help Internet service platform, retailer that are directly facing consumers can more accurately target consumers through big-data marketing[41]. And according to the individual needs of consumers, retailer can cooperate with manufacturer to produce products that meet consumer needs, avoiding product bias caused by blind guessing of consumer preferences based on small data in classic marketing, and avoiding inventory backlog caused by inaccurate market demand prediction. And all of above has been add to the article accordingly.

5.Savaskan, R.C.; Bhattacharya, S.; Wassenhove, L.N.V. Closed-loop supply chain models with product remanufacturing. Manag. Sci. 2004, 50, 239–252. [CrossRef]

6.De Giovanni, P. A joint maximization incentive in closed-loop supply chains with competing retailers: The case of spent-battery recycling. Eur. J. Oper. Res. 2018, 268: 128-147. [CrossRef]

7.De Giovanni, P.; Reddy, P. V. & Zaccour, G. Incentive strategies for an optimal recovery program in a closed-loop supply chain. Eur. J. Oper. Res. 2016, 249, 605-617. [CrossRef]

8.De Giovanni, P. State-and control-dependent incentives in a closed-loop supply chain with dynamic returns. Dyn. Gam. Appl. 2016,6, 20-54. [CrossRef]

16.Traditional CMO eliminated, Unilever establishes new "chief digital marketing officer". Mark.Sit. 2020,1,12-13.

20.JD R& D system, JD technology decryption. Beijing: Publ. Hous. Electr. Ind. 2015. [CrossRef]

41.Xiang, Z. & Xu, M. Dynamic cooperation strategies of the closed-loop supply chain involving the Internet service platform. J. Clean. Prod.2019, 220, 1180-1193. [CrossRef]

5) in games with an infinite horizon, time-variable strategies never appear, why do they occur this time?

Response: As the professor points out, in games with an infinite horizon, time-variable strategies never appear. A frequently adopted approach to dynamic optimization problems is the technique of dynamic programming. The technique was developed by Bellman (1957). The infinite-horizon autonomous problem is independent of the choice of t and dependent only upon the state at the starting time, that

is brand goodwill G in this article. Where G(t) is the state at time t along the optimal trajectory. Since

the problem depends on the current state G only, we also assume the corporate profit (optimal value

function) as a function of the state variable G. Meanwhile, since time is not explicitly involved,

the derived control variables (w, p, B) will be a function of G only.

Thank you again for your valuable suggestions, and I wish you a smooth work and a happy life.

Yours sincerely,

Ma Deqing

Round 2

Reviewer 2 Report

All my concerns were taken into account.